# Effects of Novel Photosynthetic Inhibitor [CuL_2_]Br_2_ Complex on Photosystem II Activity in Spinach

**DOI:** 10.3390/cells11172680

**Published:** 2022-08-28

**Authors:** Sergey K. Zharmukhamedov, Mehriban S. Shabanova, Margarita V. Rodionova, Irada M. Huseynova, Mehmet Sayım Karacan, Nurcan Karacan, Kübra Begüm Aşık, Vladimir D. Kreslavski, Saleh Alwasel, Suleyman I. Allakhverdiev

**Affiliations:** 1Institute of Basic Biological Problems, FRC PSCBR RAS, 142290 Pushchino, Russia; 2Bionanotechnology Laboratory, Institute of Molecular Biology and Biotechnology, Azerbaijan National Academy of Sciences, AZ1073 Baku, Azerbaijan; 3K.A. Timiryazev Institute of Plant Physiology, Russian Academy of Sciences, Botanicheskaya Street 35, 127276 Moscow, Russia; 4Department of Chemistry, Science Faculty, Gazi University, Teknikokullar, Ankara 06500, Turkey; 5College of Science, King Saud University, Riyadh 12372, Saudi Arabia; 6Department of Plant Physiology, Faculty of Biology, M.V. Lomonosov Moscow State University, Leninskie Gory 1-12, 119991 Moscow, Russia

**Keywords:** photosynthetic inhibitors, copper, metalorganic complexes, photosystem II, oxygen evolution, aromatic amino acids intrinsic fluorescence

## Abstract

The effects of the novel [CuL_2_]Br_2_ complex (L = bis{4H-1,3,5-triazino [2,1-b]benzothiazole-2-amine,4-(2-imidazole)}copper(II) bromide complex) on the photosystem II (PSII) activity of PSII membranes isolated from spinach were studied. The absence of photosynthetic oxygen evolution by PSII membranes without artificial electron acceptors, but in the presence of [CuL_2_]Br_2_, has shown that it is not able to act as a PSII electron acceptor. In the presence of artificial electron acceptors, [CuL_2_]Br_2_ inhibits photosynthetic oxygen evolution. [CuL_2_]Br_2_ also suppresses the photoinduced changes of the PSII chlorophyll fluorescence yield (F_V_) related to the photoreduction of the primary quinone electron acceptor, Q_A_. The inhibition of both characteristic PSII reactions depends on [CuL_2_]Br_2_ concentration. At all studied concentrations of [CuL_2_]Br_2_, the decrease in the F_M_ level occurs exclusively due to a decrease in Fv. [CuL_2_]Br_2_ causes neither changes in the F_0_ level nor the retardation of the photoinduced rise in F_M_, which characterizes the efficiency of the electron supply from the donor-side components to Q_A_ through the PSII reaction center (RC). Artificial electron donors (sodium ascorbate, DPC, Mn^2+^) do not cancel the inhibitory effect of [CuL_2_]Br_2_. The dependences of the inhibitory efficiency of the studied reactions of PSII on [CuL_2_]Br_2_ complex concentration practically coincide. The inhibition constant Ki is about 16 µM, and logKi is 4.8. As [CuL_2_]Br_2_ does not change the aromatic amino acids’ intrinsic fluorescence of the PSII protein components, it can be proposed that [CuL_2_]Br_2_ has no significant effect on the native state of PSII proteins. The results obtained in the present study are compared to the literature data concerning the inhibitory effects of PSII Cu(II) aqua ions and Cu(II)-organic complexes.

## 1. Introduction

Weeds, especially fast-growing and continuously emerging herbicide-resistant species, are one of the main factors limiting the yield of economically important crops. Advanced technology to increase yields through the creation of genetically modified plant species involves the simultaneous application of herbicides to kill weeds. Resistance to these herbicides has been created in cultivated plants through gene manipulation [1]. However, even with such approaches, it is necessary to search for more effective inhibitors that should act at the maximal low concentrations and should be capable of selectively suppressing photosynthesis reactions that are only characteristic of plant cells and, therefore, perhaps they could be safer for humans and fauna [2].

Photosynthesis ensures the growth and development of plants at the expense of the energy from the Sun, which is a unique characteristic of plant cells. Photosystem II (PSII), one of the main complexes of the photosynthetic apparatus (PA), splits water into electrons, protons, and oxygen [3]. Among all of the PA complexes, PSII and its oxygen-evolving complex (OEC) are the site of damages due to many stresses and agents [4], including the formation of reactive oxygen species [5]. To kill weeds without harming mammals, it is possible to suppress the activity of any PA component. However, it is more effective to suppress the most vulnerable part of PA, namely, the PSII OEC. This is how most herbicides, inhibitors of photosynthesis such as derivatives of urea, triazine, as well as phenolic inhibitors act [6].

Along with mentioned inhibitors of the photosynthetic electron transport chain, and mainly PSII, several attempts have been made to develop new inhibitory organic complexes [7,8,9,10], complexes of organic ligands with semimetals (Sb, As, etc.) [11], and transition metals (Fe, Pb, Co, Ni, Cr, Zn) [12,13,14], including copper (Cu(II))-based organometallic complexes [15,16].

Most of the used semimetals and metals (including copper cations) are practically not found in a free form in the environment due to their high ability to enter into various chemical reactions, as well as their low solubility in hydrophobic media. Copper cations (Cu^2+^) easily interact with various organic ligands, including ethylenediaminetetraacetic acid [17,18,19], and components of both organic (TRICINE, ADA, PIPES, ACES, Cholamine chloride, BES, Tris, TES, HEPES, MES, Acetamidoglycine, BICINE, Glycylglycine) and inorganic phosphate buffer solutions [20], as well as with various redox agents exogenous electron donors and acceptors, such as ascorbic acid [21], hydroxylamine [22], dithionite [23], diphenylcarbazide (DPC) [24,25], NADH [26], various quinones [27], and ferricyanide [28] widely applied in photosynthesis experiments. Therefore, in the soils, in aqueous solutions, and in various biological systems, copper is mostly present in the form of chelate organometallic complexes [29].

The complexation of a metal with an organic ligand(s) can significantly expand the possibilities of using a metal ion as an inhibitor of biological activity since it allows tuning the availability of the metal to a specific site of action by changing a number of properties of such a complex agent by selecting the optimal ligand(s), as shown, for example, for complexes of synthetic water oxidation catalysts [30,31]. The complexation of the metal with the ligand(s) endows the complex with new specific properties, facilitating the creation and maintenance of the required architecture under different conditions, increasing selectivity and the possibility of easy coordination with the target, thus enhancing the overall inhibition efficiency [32]. In the complex, not only the metal but also the ligand(s) can be biologically active, thus enhancing the overall activity of the organometallic complex. The ligand(s) may be biologically inactive but stabilize or protect the reactive metal from indiscriminate interaction with unwanted targets. In the complex based on metal that is not directly coordinated to the target, the ligand(s) can ensure the complex binds to the binding site [32]. Nevertheless, in the biological effects of organometallic complexes, the main role is still mostly played by the metal. This also applies to organometallic complexes based on copper.

Various sites and effects of Cu^2+^ aqua ions have been shown on various PSII components. Investigating the inhibitory effect of copper aqua ions on PSII, different researchers found numerous often-contradictory effects on the donor and acceptor sides, as well as on the components of the PSII reaction center. Copper Cu(II) ions replace the magnesium in the chlorophylls and bacteriochlorophyll [33]. The Cu binding site is near the OEC [34]. The inhibition of the DPIP-Hill reaction through PSII by Cu^2+^ was restored by Mn(II) but only in a phosphate (not in Hepes) buffer [35]. By suppressing the secondary electron transfer on the PSII donor and acceptor sides by incubation with linolenic acid, it was shown that copper (CuCl_2_) inhibits the primary charge separation in the PSII RC, presumably by creating damage near the RC and enhancing the dissipation of the incoming excitation energy into heat [36]. The fact that 3-(3,4-dichlorophenyl)-1,1-dimethylurea (DCMU) does not change the Cu^2+^ inhibition pattern of the room temperature fluorescence induction of chloroplasts supposes that Cu^2+^ and DCMU do not compete with each other, and consequently, have different inhibitory sites within PSII [36]. However, the opposite results [37] showed that the fluorescence intensity at room temperature decreased upon the addition of cupric sulfate and was partially restored by adding DCMU, testifying to the fact that Cu^2+^ and DCMU do compete with each other, and consequently, the Cu^2+^ inhibitory site within PSII coincides with that for DCMU, at least partially. Cu(II) has a specific inhibitory binding site by interacting with an essential amino acid group(s) that can be protonated or deprotonated at the inhibitory binding site, and the Cu(II) inhibitory mechanism is non-competitive with respect to 2,6-dichloro-1,4-benzoquinone (DCBQ) and DCMU and competitive with respect to H^+^ [38]. Cu influences the PSII electron transport on the acceptor side of the Pheo-Fe-Q_A_ domain [39,40]. CuCl_2_ greatly reduces the binding affinity of atrazine for PSII (K_D_) but does not change the number of binding sites compared to the control [41]. Furthermore, 100 μM CuSO_4_ does not affect the charge separation and the formation of (P_680_^+^Q_A_^−^) but changes the properties of TyrZ and/or its immediate microenvironment, thus blocking the electron transfer to P_680_^+^ [42]. Cu^2+^ (CuSO_4_) inhibits TyrZ oxidation by P_680_^+^, eliminating the light-induced EPR signal II rapidly, and the effect cannot be reversed by EDTA [43]. On PSII membranes, when using picosecond time-resolved fluorescence, it was shown that Cu(II) does not affect primary radical pair formation but strongly affected the formation of a relaxed radical pair by neutralizing the negative charge on Q_A_^−^ and eliminating the repulsive interaction between Pheo^−^ and Q_A_^−^ and/or by modifying the general dielectric properties of the protein region surrounding these cofactors. Furthermore, a close attractive interaction between Pheo^−^, Q_A_^−^, and Cu^2+^ has been supposed [44]. In a PSII membrane, Cu enhances photoinhibition [45], catalyzing the production of hydroxyl radicals [45]. Ca^2+^ competitively prevents Cu^2+^ inhibition of O_2_ evolution activity in NaCl-washed PSII membranes [46]. As part of a Cu-based complex, Cu substitutes endogenous Ca^2+^ and Mn^2+^ ions from the OEC [15]. Cu ions of the compounds studied in [15] and other aqua Cu-based complexes [47] affect both the TyrZ of D1 and the TyrD of D2 proteins of the PSII reaction center. CuCl_2_ oxidizes both forms (LP and HP) of cytochrome b_559_, at low (1–10 μM) and higher (10–100 μM) concentrations, respectively, as well as chlorophyll Z, and these sites are a source of copper-induced fluorescence quenching and the inhibition of oxygen evolution in PSII [48]. Comprehensive reviews of the different sites and effects of Cu^2+^ aqua ions on various PSII components are detailed in several publications [49,50]. The aim of this work was to investigate the possible effects of a novel Cu(II) organometallic complex (bis{4H-1,3,5-triazino [2,1-b]benzothiazole-2-amine,4-(2-imidazole)}copper(II) bromide complex) (CuL_2_]Br_2_) on photosystem II. To analyze the effects of this complex on the activity of PSII membranes isolated from spinach, we studied photosynthetic oxygen evolution by PSII membranes; the photoinduced changes of the PSII chlorophyll fluorescence yield (Fv) related to the photoreduction of the primary quinone electron acceptor, Q_A_; as well as intrinsic fluorescence of aromatic amino acids (components of the PSII proteins); at different [CuL_2_]Br_2_ concentrations. The results obtained in the present study are compared to the literature data on the inhibitory effects of PSII Cu(II) aqua ions and Cu(II)-organic complexes.

## 2. Materials and Methods

### 2.1. Thylakoids Isolation

Thylakoids were isolated from the leaves of the greenhouse spinach (*Spinacia oleracea* L.) according to a method similar to the one described previously [51], and were suspended in a medium (A) containing 20 mM Mes–NaOH (pH 6.5), 0.33 M of sucrose, 15 mM of NaCl, 10 mM of MgCl_2_, and 10% glycerol and kept at −80 °C.

### 2.2. PSII Membranes Isolation

The oxygen-evolving PSII-enriched thylakoid membrane fragments (PSII membranes) were isolated from the leaves of the greenhouse spinach (*Spinacia oleracea* L.), as described earlier [52], with a little modification as in [53]. A similar PSII membrane ratio of RC (determined from pheophytin photoreduction)-to-chlorophyll molecules is about 1/200–220 was used [54]. The membranes were capable of evolving photosynthetic oxygen with rates of 400–500 μmol O_2_ (mg Chl)^−1^ h^−1^ under saturating light in the presence of 0.1 mM 2,5-dichloro-p-benzoquinone and 1 mM K_3_Fe(CN)_6_ as electron acceptors. The PSII membranes were suspended in a medium (A) and were fast frozen for storage at −80 °C. The total chlorophyll concentration of the PSII membranes was determined using 96% (*v*/*v*) ethanol, as described earlier [55].

### 2.3. Photosynthetic Oxygen Evolution Measurements

The rate of steady-state light-induced oxygen evolution using the PSII membranes was measured polarographically in Medium A under continuous saturating red light (λ > 650 nm, 1200 μmol photon s^−1^ m^−2^) at 25 °C using a Clark-type Hansatech oxygen electrode (Hansatech Instruments Ltd., Norfolk, UK) connected to a water-temperature-controlled thermostat in the presence of a combination of two artificial electron acceptors: 100 µM of a strongly lipophilic dichloro-p-benzoquinone (DCBQ) dissolved in dimethyl sulfoxide (DMSO) and 1 mM potassium ferricyanide (FeCN) as additional electron acceptors to keep DCBQ in an oxidized state [56]. The PSII membrane concentration was equivalent to 20 µg mL^–1^ of chlorophyll. The rates were estimated from the slope of the kinetics during the first 30 s after turning on the actinic light. Previously [57], according to the data on the dependence of the photoinduced oxygen evolution rate by PSII particles upon DCBQ concentration, it was shown that the highest oxygen yields were recorded in the presence of 25–200 µM DCBQ.

### 2.4. Photoinduced Changes of the PSII Chlorophyll Fluorescence Yield Measurements

The photochemical activity of the PSII was also analyzed by measuring the kinetics of the light-induced changes of the PSII chlorophyll fluorescence yield (ΔF), related to the photoreduction of the PSII primary electron acceptor, plastoquinone Q_A_, with a pulse-amplitude modulation (PAM) fluorometer (XE-PAM, Heinz Walz, Germany) and accompanying software Power Graph Professional 3.3 in Medium A in a 10-mm cuvette at room temperature. The characteristic values F_V_, F_0_, F_M_, and the maximum quantum photochemical yield of PSII (ratio F_V_/F_M_), as well as the ratio F_V_/F_0_, were determined. Here, F_0_ is the level of fluorescence before actinic light is applied, measured under weak measuring light illumination, and F_M_ is the maximum fluorescence level, F_V_ = F_M_ − F_0_. To record the F_0_ level of fluorescence, the preliminary dark-adapted sample was illuminated using weak probe pulses of measuring light (λ = 490 nm; 4 µmol m^–2^ s^–1^; Xenon-Measuring Flash Lamp, 64 Hz, BG39, Schott). For the registration of the photoinduced changes in PSII chlorophyll fluorescence yield (F_V_), the dark-adapted samples were illuminated by the light of saturating intensity (λ = 490 nm; 1000 µmol m^–2^ s^–1^, BG39, Schott).

### 2.5. Interaction between PSII and PSI through the Electron Transport Chain Measurements

The effect of the [CuL_2_]Br_2_ complex on the interaction between PSII and PSI through the electron transport chain was studied on thylakoids by a characteristic decrease in the yield of PSII chlorophyll fluorescence due to the oxidation of Q_A_^−^ upon the additional excitation of PSII by the so-called Light 1, which predominantly excites PSII, as in [58]. Light 1 was obtained by combining two light glass filters, SZS20 and KS19. The transmission spectrum of the light filters combining (Light 1) is shown on the corresponding Figure lower in paragraph 3.7. The sequence of actions was as follows. After the F_M_ level was reached (Q_A_ was completely reduced) as a result of the illumination of the thylakoids, with Light 2 predominantly exciting PSII, Light 2 was switched off, and during the dark decline of F_M_ (dark oxidation of Q_A_^−^), Light 1 was switched on. In this case, when the interaction between PSII and PSI (control) is not inhibited, an acceleration of the dark decay of F_M_ (oxidation of Q_A_^−^) was observed due to the additional predominant excitation of PSI by Light 1.

### 2.6. Aromatic Amino Acids Fluorescence Measurements

Possible changes in the state, structure, and the environment of protein(s) (including PSII proteins) as a result of the binding caused by an inhibitory agent can be detected by changes in the intrinsic fluorescence (mainly fluorescence intensity changes) of amino acids with aromatic ring side chains (tryptophan (Trp), tyrosine (Tyr), and phenylalanine (Phe)), predominantly Trp and Tyr [8,9,59]. The probable interaction of [CuL_2_]Br_2_ with PSII membrane proteins was investigated by monitoring the changes in the intrinsic fluorescence of the aromatic amino acids of these proteins according to the methods described previously in [8,9,59]. The fluorescence emission spectra of the aromatic amino acids suspended in the PSII membranes were recorded on an Agilent Cary Eclipse fluorescence spectrophotometer (Agilent Technologies, Inc., Santa Clara, CA, USA). The samples of the PSII membrane suspension with and without the studied complex, [CuL_2_]Br_2_, were excited at a wavelength of 275 nm, with an excitation spectral slit width of 20 nm, and fluorescence emission was registered in a range from 290 nm to 420 nm, with an emission spectral slit width of 10 nm. Before the measurements, the samples were incubated in the presence of the studied complex in the dark for 3 min (or 21 min). The complex was added to the PSII membrane suspension in the DMSO solution. The DMSO concentration in all of the samples was the same as in the control (not more than 5%). The PSII–membrane concentration was equivalent to 10 µg mL^−1^ of chlorophyll.

## 3. Results

### 3.1. Synthesis of Ligand and the Copper(II) Complex [CuL_2_]Br_2_

#### 3.1.1. Synthesis of Ligand

Ligand 4H-1,3,5-triazino [2,1-b]benzothiazole-2-amine,4-(2-imidazole) was synthesized for the first time by a condensation reaction between aldehydes (2-thiophenecarbaldehyde/2-imidazolecarbaldehyde) and 2-guanidobenzimidazole or 2-benzothiazoleguanidine. A solution of 2-benzothiazoleguanidine (0.192 g) in methanol (25 mL) was treated with 2-imidazolcarboxylaldehyde (0.096 g) and two drops of piperidine. The mixture was stirred at 60 °C for 5 days. The resulting orange solid was separated by filtration, washed with methanol, and dried under a vacuum. Calculated for C_12_H_10_N_6_S (mol. wt. 270.31): C, 53.32; H, 3.73; N, 31.09; S, 11.86. Found: C, 53.39; H, 3.79; N, 31.11; S, 11.89. IR (KBr, $\bar{\nu }$/cm^−1^): 3412, 3296, 1639, 1509, 1249. LC-MS: *m*/*z* = 270.31 (M^+^). ^1^H-NMR(400 MHz, DMSO-d_6_) δ: 6.65 (1H, s)

#### 3.1.2. Electrochemical Synthesis of the Copper(II) Complex [CuL_2_]Br_2_

The new copper(II) complexes [CuL_2_]Br_2_ were synthesized using an electrochemical method [60], using a Pt wire as the cathode (1 cm × 1 cm) and a copper plate as the sacrificial anode. Thus, a methanol solution (30 mL) of the ligand (~1 mmol) containing about 15 mg of tetrabutylammonium bromide as the supporting electrolyte in a beaker (100 mL) was electrolyzed at room temperature for 3 h. An applied voltage of 5 V allowed for sufficient current flow for the smooth dissolution of the copper. The obtained solid was isolated, washed with water, and dried under a vacuum. The structure of the tested ligand (4H-1,3,5-triazino [2,1-b]benzothiazole-2-amine,4-(2-imidazole)) and Cu(II)-complex (Bis{4H-1,3,5-triazino [2,1-b]benzothiazole-2-amine,4-(2-imidazole)}copper(II) bromide) is shown in Figure 1A,B, the expanded structural formulas; C, the optimized molecular structures. C_24_H_22_Br_2_CuN_10_: IR (KBr, $\bar{\nu }$/cm^−1^):3402, 3251, 1600, 1552, 1263. LC-MS: *m*/*z* = 566.58 (M^+2^). Elemental analysis: C:39,49; H:3,04; Br:21,90; Cu:8,71; N:26,87.

### 3.2. Effects of [CuL_2_]Br_2_ on the Photosynthetic Oxygen Evolution

Because the aqua ions of Cu(II) (1) can oxidize many inorganic and organic compounds (shown upper) (2), stimulate photosynthetic oxygen evolution at low concentrations [61,62], and (3) possible aqua ions of Cu(II) saved these abilities as part of [CuL_2_]Br_2_, we tested [CuL_2_]Br_2_ as an artificial electron acceptor for PSII in a photosynthetic oxygen evolution reaction. However, in the presence of only [CuL_2_]Br_2_ (from 1 µM up to 30 µM) in the absence of other artificial electron acceptors (AEA), no oxygen evolution by PSII membranes was observed, as seen in Figure 2 (kinetics 6).

In the presence of AEA (DCBQ plus FeCy) without [CuL_2_]Br_2_ (control), the PSII membranes generate oxygen under continuous illumination with saturating light at a rate of about 480 μmol O_2_ (mg Chl h)^−1^ (Figure 2, kinetics 1). This activity is comparable to that described earlier for similar PSII membranes [63,64]. [CuL_2_]Br_2_ inhibits oxygen evolution in a concentration-dependent manner. At concentrations of 3 µM (2), 20 µM (3), 50 µM (4), and 100 µM the reaction rate is decreased by about 15%, 43%, 59%, and 69% versus the control (100%) (Figure 2, kinetics 2–5). The reaction is suppressed almost completely by 10 mM [CuL_2_]Br_2_ (not shown). The degree of PSII membrane inhibition does not depend on the duration of the incubation (5–20 min) in the presence of [CuL_2_]Br_2_.

### 3.3. Effects of [CuL_2_]Br_2_ on the Photoinduced Changes of the PSII Chlorophyll Fluorescence Yield

The suppression of photosynthetic oxygen evolution is one of the most significant indications that the present agent is indeed an inhibitor of PSII functional activity. However, the results obtained in our study do not make it possible to judge whether (1) this agent acts on the donor or acceptor side or at the RC level, or (2) which intermediate of the PSII electron transport chain is affected by this agent, and (3) whether this agent acts as an electron transfer inhibitor or as an exogenous electron donor (acceptor) capable of effectively competing with native endogenous components and thus disrupting the normal course of PSII photochemical reactions. Information (a) on the localization of the inhibition site of this inhibitor in PSII, (b) on the number of sites of action and location of the main and/or additional site(s) of inhibition, and (c) on the possible mechanism of action of the inhibitor can be obtained using the photoinduced changes of the PSII chlorophyll fluorescence yield (F_V_) related to the photoreduction of the primary quinone electron acceptor, Q_A_, including the using the exogenous electron donors (acceptors) and/or known inhibitors with well-studied effects on PSII. Therefore, we decided to use this method in order to obtain more detailed information about the site(s) and mechanism of the inhibitory effect of [CuL_2_]Br_2_ on PSII. The results of these studies are presented in Figure 3.

Figure 3 shows the kinetics of the photoinduced changes of the PSII chlorophyll fluorescence yield (F_V_) related to the photoreduction of the primary quinone electron acceptor, Q_A_, in the PSII sub-chloroplast membrane particles in the absence of any additions (kinetics 1) and in the presence of [CuL_2_]Br_2_ at concentrations of 8 µM (kinetics 2), 30 µM (kinetics 3), and 100 µM (kinetics 4). In these experiments, we determined the values of F_0_ and F_M_ as the maximum and minimum fluorescence yield of chlorophyll “a” in the dark-adapted samples. Correspondingly, F_V_ calculated as F_M_–F_0_, the ratio of Fv/F_M_, recorded the possible changes in these parameters caused by the action of the inhibitor compared to the control. The control kinetics (measured in the absence of an effector) had an F_V_/F_M_ ratio of 0.79 ± 0.02. This value indicates a rather high photochemical activity of PSII in the used PSII membranes. In the presence of [CuL_2_]Br_2_, a significant decrease in the F_M_ level occurs (kinetics 2), and this decrease in the F_M_ level is greater at higher concentrations of added [CuL_2_]Br_2_ (kinetics 3 and 4). It is very important to note here that at all of the studied concentrations of [CuL_2_]Br_2_, the decrease in the F_M_ level occurs exclusively due to a decrease in F_V_; [CuL_2_]Br_2_ does not cause any changes in the F_0_ level (neither increase nor decrease) (kinetics 2–4). Therefore, to estimate [CuL_2_]Br_2_ inhibitory efficiency, we chose F_V_ as a percentage relative to the control. The inhibition of PSII photoinduced F_V_ is 66%, 48%, and 32% at [CuL_2_]Br_2_ concentrations of 8 µM (kinetics 2), 30 µM (kinetics 3), and 100 µM (kinetics 4), respectively. These data are in good agreement with the data obtained in experiments on photosynthetic oxygen evolution (Figure 2). It is also necessary to note one more important experimental fact that, in the presence of all of the studied concentrations of [CuL_2_]Br_2_, there is (a) both slowing down and acceleration of the dark relaxation of F_M_, reflecting that the dark reoxidation of the reduced Q_A_ occurs; (b) no retardation of the photoinduced rise in F_M_ also occurs. Exogenous electron donors (sodium ascorbate, DPC, Mn^2+^) do not remove the inhibitory effect of [CuL_2_]Br_2_ on F_V_, regardless of the sequence of addition of these reagents (not shown). Moreover, the degree of F_V_ suppression by the same concentration of [CuL_2_]Br_2_ does not increase with increasing incubation time (3 and 21 min) in the presence of this inhibitor in the dark (not shown).

### 3.4. Dependence of the PSII Membranes Photochemical Activity Inhibition on [CuL_2_]Br_2_ Concentration

Figure 4 shows the dependence of the PSII membranes photochemical activity inhibition on [CuL_2_]Br_2_ concentration, measured as both photosynthetic oxygen evolution (black squares) and as light-induced changes in the PSII chlorophyll fluorescence yield (red circles). Both of the presented dependences were measured at a PSII membranes concentration equivalent to 20 µg mL^–1^ of chlorophyll. The dependence of the oxygen evolution inhibition almost completely coincides with that of the suppression of the photoinduced changes in the PSII chlorophyll fluorescence yields over the entire range of the investigated agent concentrations. Usually, a special indicator is used to quantitatively assess inhibitor potency. It is IC_50_ (or pIC_50_—the IC_50_ value converted to a negative decadic logarithm of IC_50_ (-logIC_50_) for unification)—the concentration of agent needed to inhibit the investigated activity to half that of the initial value (control) [65,66]. IC_50_ is an operational parameter to assess the effective strength of a particular inhibitory compound, and it is strongly dependent on PSII membrane concentration, and is usually expressed as the concentration of chlorophyll [67,68]. In 1972 Cedeno-Maldonado et al. [69] also showed that the inhibition of uncoupled electron transport in chloroplasts by copper ions Cu^2+^ occurs at the PSII level and depends on the inhibitor/chlorophyll ratio. At high concentrations of chlorophyll, more Cu^2+^ ions are required to achieve the same degree of inhibition [69].

As can be seen from the data presented in Figure 4, for both of the measured characteristics of the PSII photochemical reactions, the values of the [CuL_2_]Br_2_ concentrations induce the suppression of half of the maximal studied activities (IC_50_), are about 30 µM. [CuL_2_]Br_2_ in concentrations of a little bit more than 0.1 µM only induces a very weak suppression of these PSII photochemical activities. An almost complete (more than 90% versus control) suppression of both reactions was observed at an agent concentration of 10 mM. It should be noted that all of the presented data were obtained at a concentration of PSII membrane of 20 µg mL^−1^ chlorophyll.

### 3.5. Graphic Determination of Ki Value

Because IC_50_ depends on PSII membranes concentration, it cannot be used to compare the results obtained from different inhibitors or for the same inhibitors obtained by various scientific groups without specifying the target concentration for which that particular IC_50_ value was determined. In contrast to IC_50_, another parameter, Ki (the inhibition constant characterizing the intrinsic measure of binding affinity between the inhibitor and its target site), does not depend on the concentration of the PSII membranes. It is a known method for determining Ki based on the IC_50_ values data obtained for different concentrations of the target (in our case, PSII membranes) [67]. We also applied this method (Figure 5). We determined the IC_50_ values for the suppression of light-induced oxygen evolution by [CuL_2_]Br_2_ for nine different concentrations of PSII membranes. We then plotted IC_50_ as a function of chlorophyll concentration as a line plot and extrapolated the resulting line to a zero chlorophyll concentration, as described previously [67]. The resulting IC_50_ value that corresponds to a zero chlorophyll concentration is Ki (Figure 5). In our case, Ki is about 16 µM (16.2 ± 0.2), and –logKi is 4.8. Based on the explanation and equation (I_50_ = Ki + 1/2 xt) given by [67], it is possible to estimate the concentration of the [CuL_2_]Br_2_-specific binding sites (xt) to the chlorophyll concentration. In the case of the PSII membrane concentrations used by us, for example, equal to 20 μg ml^−1^ or about 22 μM, we determined that xt equals 1, i.e., one [CuL_2_]Br_2_ molecule on one chlorophyll molecule.

### 3.6. Graphic Estimation of the [CuL_2_]Br_2_ Binding Sites Number

It is known [68] that if the available inhibition data present a so-called Hill plot (dependence of log(inhibition/(1-inhibition)) on the log (inhibitor concentration), then it will be possible to estimate the number of binding sites on the acceptor molecule based on the slope of the obtained line. We used this method for the inhibition data of the PSII light-induced oxygen evolution by various [CuL_2_]Br_2_ concentrations. Figure 6 shows such a Hill plot of these data. As follows from Figure 6, the slope of the obtained line is about 1 (0.87 ± 0.03).

### 3.7. Effects of [CuL_2_]Br_2_ on Interaction between PSII and PSI through the Electron Transport Chain Measurements

A significant number of known inhibitors that affect PSII, act on its acceptor side, blocking the re-oxidation of reduced Q_A_ by PSI [70]. In addition to the main site of action, many of them have at least one other additional site of action. The effects of the inhibitor on the main and additional sites of inhibition are observed at different concentrations of the inhibitor. The inhibition at the additional site occurs at higher concentrations of the suppressor [70]. Given the above information, we decided to test how [CuL_2_]Br_2_ affects the efficiency of light-induced oxidation by PSI of pre-reduced Q_A_.

The light-induced interaction of PSI with PSII is possible to estimate when measuring the dark relaxation of the photoinduced changes in the PSII chlorophyll fluorescence yield (variable F) related to the dark oxidation of the reduced Q_A_ (Q_A_^−^) from the F_M_-level to level F_0_ and to switch it on at a determined moment of actinic Light 1 (L1), predominantly exciting PSI (λ > 700 nm). The transmission spectrum of Light 1 is shown in Figure 7B. In this case, induced by Light 1, the acceleration of Q_A_^−^ re-oxidation is observed. For comparison, we used DCMU (3-(3,4-dichlorophenyl)-1,1-dimethylurea) and DBMIB (2,5-dibromo-3-methyl-6-isopropyl-p-benzoquinone). DCMU and DBMIB are available, well studied, and the most often and ubiquitously used in scientific research inhibitors. The mechanisms of the inhibitory actions of these agents are well studied. DCMU blocks Q_A_^−^, the oxidation by plastoquinone (PQ) from the membrane pool by binding instead of PQ with the Q_B_-plastoquinone-herbicide-binding protein (Q_B_-site of D1-protein) of the PSII reaction center [70]. DBMIB blocks oxidation driven by PSI through the cytochrome (cyt) b6f of plastoquinol (PQH2) from the membrane PQ pool [70]. DBMIB binds to the Qo (Qp) site of the cytochrome b6f complex or very close to it [70,71].

The results of such experiments on thylakoids are shown in Figure 7A. In the absence of any inhibitors (control, kinetics 1), actinic Light 1 induced an appreciable acceleration of Q_A_^−^ oxidation. In the presence of DCMU (0.5 µM), actinic Light 1 did not induce the acceleration of Q_A_^−^ oxidation. Instead of this, Light 1 induced an increase in PSII chlorophyll fluorescence (kinetics 2). This does not indicate the oxidation of Q_A_^−^, but, on the contrary, the reduction of those Q_A_ molecules that had time to oxidize in the process of the dark relaxation of fluorescence before the moment actinic Light 1 was turned on. Why is this occurring? Light 1 excites not only PSI but also PSII, but to a much lesser extent because Light 1 contains a negligible fraction of the light that is absorbed by the main PSII pigments (Figure 7B). On the other hand, it has been shown that the far-red light limits of PSII photochemistry in the PSII membranes from spinach expend up to 800 nm, but this light has a substantially decreased efficiency for PSII [72]. In our case (when Q_A_^−^ oxidation is blocked by DCMU), the substantially decreased efficiency of PSII actinic Light 1 is enough to induce the photoaccumulation of reduced Q_A_, and it is revealed as an increase in the PSII chlorophyll fluorescence yield. The effect of DBMIB (7 µM) on the interaction of PSI with PSII, revealed by the changes in the PSII chlorophyll fluorescence yield, is practically similar to the effect of DCMU (Figure 7A kinetics 3). As in the case of DCMU, the presence of DBMIB Light 1 induces an increase in PSII chlorophyll fluorescence yield, but in this case, the increase in fluorescence occurs at a slightly slower rate than in the case of DCMU. These comparative experiments with DCMU and DBMIB showed that the appearance of the increase in the PSII chlorophyll fluorescence yield under Light 1 illumination during the dark relaxation of F_M_ is a reliable pointer to the fact that Q_A_^−^ oxidation by PSI was inhibited at least at the Q_B_^−^ or Qo site and we could use the pointer to check the [CuL_2_]Br_2_ results. The investigations of the [CuL_2_]Br_2_ effect on the interaction of PSI with PSII are shown in Figure 7A kinetics 4. As can be seen in this figure, no increase in the PSII chlorophyll fluorescence yield occurs in response to the illumination with actinic Light 1. There is only a very slight slowdown in the dark fluorescence decay compared to the control. It should be noted that this effect (very slight slowdown of the florescence dark decay) of [CuL_2_]Br_2_ is observed when F_M_ is already suppressed by more than 50%.

### 3.8. Effects of [CuL_2_]Br_2_ on the Aromatic Amino Acids Intrinsic Fluorescence

Many organic and inorganic substances can affect the structural and functional properties of proteins. Damage and/or changes in the native structure of the proteins that make up photosystem complexes or the properties of their surrounding environment may be one of the causes of disturbances in the photosynthetic electron transfer and the inhibition of photosynthesis in general [8,9,10,13]. Heavy metal ions and their complexes, including Cu ions and Cu complexes, can also act in a similar way. Copper(II) has been shown to inhibit photosynthetic oxygen evolution by binding to a non-protonated protein residue of a component very close to the PSII water-splitting system [73]. At the same time, such changes in PSII proteins can be caused both by the action of metal ions in the composition of the organometallic complex (Cu) [15] and by the action of the ligand, which is the organic part of such a complex [8,9,10,13].

Three amino acids with aromatic circular side chains—phenylalanine, tyrosine, and especially tryptophan—are sensitive to the local environment in the proteins, and these AAA undergo changes in the wavelength and/or intensity of their intrinsic fluorescence upon any impact on the protein [74]. One of the most accessible, and at the same time, quite reliable and informative methods for detecting possible similar effects on photosystem proteins, is the method of recording the intrinsic fluorescence of aromatic amino acids (AAA) that make up proteins, but mainly the intrinsic tryptophan and tyrosine fluorescence [59]. It was shown that some of the previously studied organic [8,9,10] and organometallic complexes [15], as well as free metal cations [13], which have an inhibitory effect on the photochemical activity of PSII, also caused a significant concentration-dependent decrease in the intensity of the intrinsic fluorescence of the PSII protein aromatic amino acids. It could be assumed that some negative effect(s) of the studied [CuL_2_]Br_2_ on the native state and/or structure of the PSII protein(s) is the reason for, or at least contribute to, the suppression of photosynthetic oxygen evolution and the magnitude of the photoinduced changes in PSII chlorophyll fluorescence yield. Therefore, we decided to test whether [CuL_2_]Br_2_ affects the structure of the PSII proteins by comparing the parameters of the AAA intrinsic fluorescence in the suspension of the PSII membranes without and with [CuL_2_]Br_2_. The results of these studies are presented in Figure 8.

Figure 8 shows the fluorescence emission spectra of the AAA of the untreated PSII membranes (spectrum 1) and the PSII membranes treated with [CuL_2_]Br_2_ at various times (spectra 2, 3). Upon excitation with light at a wavelength of 275 nm, a broad band was observed in the emission spectrum with a broad maximum at 330 nm and a shoulder at 350 nm. This is the emission of the intrinsic fluorescence of tyrosine, tryptophan, and phenylalanine [15,75]. Since this fluorescent emission is sensitive to changes in the local environment of the above aromatic amino acids, and if we assume that [CuL_2_]Br_2_ has any effect on PSII proteins, then we could expect any changes in the intensity and/or any shift in the maxima of this fluorescence if the native configuration of the PSII proteins will be disturbed due to the action of [CuL_2_]Br_2_ or as a result of binding to the PSII proteins of the [CuL_2_]Br_2_ complex or the Cu2+ ions released from [CuL_2_]Br_2_. It is known that, in parallel with a decrease in the intensity of the intrinsic fluorescence of AAA in the protein, a shift in the F maximum is also possible. This indicates a change in the fluorophore microenvironment as a result of the effector impact [76].

However, in all of the cases (3 min and 21 min incubation of the PSII membranes with [CuL_2_]Br_2_ at a concentration of 30 μM), the intensity of the intrinsic fluorescent emission of PSII proteins did not decrease in the presence of [CuL_2_]Br_2_ at any of the indicated wavelengths (Figure 8 spectra 2, 3). Furthermore, no pronounced shifts of these emission maxima or changes in the shape of these spectra were observed (Figure 8 spectra 2, 3). This indicates the absence of any effect of [CuL_2_]Br_2_ on the native structure of PSII proteins or the absence of [CuL_2_]Br_2_ binding to any PSII protein. In addition, no new fluorescence bands were observed in this wavelength range, which could indicate the possible formation of a complex between [CuL_2_]Br_2_ and the PSII proteins. Similar results were obtained in the presence of [CuL_2_]Br_2_ at a concentration of 90 μM, inducing an almost complete suppression of the photochemical activity of PSII (not shown).

## 4. Discussion

As there is no photosynthetic oxygen evolution due to the PSII membranes observed in the absence of artificial electron acceptors, but in the presence of the studied agent at a concentration of 0.1 mM (Figure 2, kinetics 6), it can be concluded that the studied agent [CuL_2_]Br_2_ (or, for example, its putative oxidized form), obviously, is not able to act as an exogenous PSII electron acceptor. These results could be taken as an indication that the copper cation in the complex is sufficiently strongly bound to the ligand, and the charge on the copper cation is balanced by a total charge on the ligand sufficient to prevent the possible interaction of the copper cation with other components of the environment as electron acceptor. The previously observed stimulation of the photosynthetic oxygen evolution by the Cu^2+^ at low concentrations [61,62] could be due to Cu^2+^ functioning as an uncoupler at this concentration, as was proposed earlier [35] based on the stimulation of the overall electron transfer rate by low Cu^2+^ concentration at least in the case of *Lemna* fronds [61]. In thylakoids (closed membrane vesicles–organelles of chloroplasts, plant cells), electron transfer is coupled with proton transfer from outside (stroma) inside the thylakoid (lumen) with the formation of a proton gradient. Uncouplers disturb the conjugation between electron transfer and proton gradient formation, and all of the energy is only used for electron transport. The PSII membranes are open grana membrane fragments and not whole-membrane vesicles. Therefore, in PSII membranes, a proton gradient does not form. Consequently, uncouplers do not affect the efficiency of electron transfer in PSII membranes, as this one has a place in thylakoids. The decrease in the photosynthetic oxygen evolution rate in the PSII membranes in the presence of exogenous electron acceptors is observed if [CuL_2_]Br_2_ is present in the measurement medium; it may indicate that [CuL_2_]Br_2_ inhibits the functional activity of some PSII electron transport chain components. However, it can also be assumed that this observed effect is the result of a direct chemical interaction between AEA-s and [CuL_2_]Br_2_ (for example, oxidation of AEA-s by [CuL_2_]Br_2_) leading to the depletion of AEA-s concentrations and, as a consequence, to the decrease in the rate of oxygen evolution. It can be assumed that such an AEA in the used pair of AEA-s can be each of these agents or one of them: DCBQ or FeCN. If we compare the potentials of substances oxidized by the Cu(II) cation: ascorbate E_m,7_ = +60 mV [77]; 1,5-diphenylcarbazide E_m,7_ = −300 mV [78]; glutathione E_m,7_ = −240 mV [79]; NADH E_m,7_ = −320 mV [77]; DCBQ E_m,7_ = +309 mV [80] and FeCN E_m,7_ = +420 mV [81], it is obvious that such an AEA can be DCBQ and not FeCN. On the other hand, it was shown that the redox potential of Cu(II) in the complex could differ from the reduction potential of the Cu(II) Cu(I) aqua couple (164 mV) [82] and be significantly more positive [83], for example, including a number of copper proteins: Rusticyanin (bacteria) 680 mV; Plastocyanin (plants) 370 mV; Amicyanin (bacteria) 294 mV; Stellacyanin (plants) 285 mV; and Azurin (bacteria) 276 mV [84]. Taking these data into account, it can be assumed that FeCN can also be depleted due to direct chemical interaction with [CuL_2_]Br_2_. However, the following experimental facts testify against this assumption: (1) The absence of changes in the absorption spectra of both FeCN and [CuL_2_]Br_2_ upon mixing and co-incubation for a sufficient time under the conditions of measuring the photosynthetic oxygen evolution rate but in the absence of the PSII membranes (not shown); (2) [CuL_2_]Br_2_ does not have any absorption bands in the red region of the spectrum (λ > 650 nm) (not shown). Therefore, a [CuL_2_]Br_2_ concentration-dependent decrease in the rate of the photosynthetic oxygen evolution is not result of the actinic light screening by [CuL_2_]Br_2_. The hypothesis that the decrease in the photosynthetic oxygen evolution by PSII membranes is a consequence of the direct interaction of [CuL_2_]Br_2_ with DCBQ or FeCN could be tested by increasing the concentration of FeCN and/or DCBQ. If the inhibition efficiency did not change, then this would testify against the above proposal. However, the concentration of FeCN is already quite high (1 mM), above which it is quite possible to expect non-specific effects of FeCN on the PSII-membranes. On the other hand, it is not recommended to use DCBQ at concentrations above 250–300 µM. Based on the experimental data, it is proposed that DCBQ taking an electron from the reduced species in PSII may then donate it to P_680_^+^˙ or Tyr_Z_˙, while ferricyanide prevents this process through the oxidation of the reduced DCBQ [85]. Therefore, the addition of FeCN is required to prevent the accumulation of reduced DCBQ [56,86]. At the same time, other authors, for example [38], measured the oxygen evolution activity of PSII membranes (500 µmol O_2_ mg Chl^−1^ h^−l^) in the presence of DCBQ (without FeCN). It can be assumed that [CuL_2_]Br_2_ acts as an exogenous electron donor capable of competing with the endogenous OEC for positive equivalents from RC and, therefore, for the ability to supply electrons for the reduction of the DCBQ + FeCN pair, not directly as a result of a direct chemical reaction, but through the components of RC. This could be manifested as a decrease in the rate of light-induced oxygen evolution with an increase in the concentration of [CuL_2_]Br_2_. However, the above indirect data on the redox properties of Cu^2+^ aqua ions and the effect of organic ligands on them makes this assumption unlikely. The fact that the degree of inhibition of the photosynthetic oxygen evolution rate by PSII membranes does not depend on the time of incubation in the presence of [CuL_2_]Br_2_ in the dark (effect measured after 3 min is equal to that after 21 min) indicates that [CuL_2_]Br_2_ sufficiently reaches the target quickly, apparently localized in the hydrophobic region of the PSII pigment–protein complex. This assumption is also supported by the fact that [CuL_2_]Br_2_ is very poorly soluble in an aqueous media and, therefore, a stock solution of [CuL_2_]Br_2_ was prepared in DMSO. Based on the above reasoning, we are inclined to assume that the decrease in the photosynthetic oxygen evolution rate by PSII membranes observed in the presence of [CuL_2_]Br_2_ is the result of inhibition by [CuL_2_]Br_2_ of the functional activity of some component of the PSII electron transport chain. Earlier [87], it was shown that PSII oxygen evolution inhibition induced by Cu(II) is reversible in the presence of a Cu(II)-chelating agent (EDTA), but the reversibility is strongly dependent on the incubation time with Cu(II) in light. Only 30 s of illumination in the presence of Cu(II) is enough for the inhibition of PSII by the cation and would be totally irreversible by EDTA.

The PSII maximum quantum photochemical yield of the dark-adapted samples φPo = Fv/F_M_ being equal to 0.79 ± 0.02 testifies to the fact that the photochemical activity of PSII in the used PSII membranes is rather high, and these PSII membranes can be used to explore the [CuL_2_]Br_2_ inhibitory effects on PSII. The effective decrease in F_M_ caused by [CuL_2_]Br_2_ may be due to the fact that (a) no separation and stabilization of charges in the RC occurs; (b) due to the inhibition of the functioning of some components of the OEC, electrons from the OEC through the RC to Q_A_ are not supplied, which are necessary for the photoaccumulation of the reduced Q_A_ and, therefore, F_M_ cannot reach a level as in the control; (c) [CuL_2_]Br_2_ acts as an exogenous electron acceptor that removes an electron from Q_A_ more efficiently than the OEC supplies electrons to Q_A_ and therefore photoaccumulation of the reduced Q_A_ does not occur, and it is revealed as a decrease in F_M_. (c) [CuL_2_]Br_2_ is a quencher of the excited states of chlorophyll, and this manifests itself as the quenching of chlorophyll fluorescence. Let us consider these assumptions in reverse order.

(d) [CuL_2_]Br_2_ only selectively reduces the intensity of the variable chlorophyll fluorescence (Fv) and does not affect the level of F_0_ (Figure 3). Taking into account the fact that [CuL_2_]Br_2_ does not decrease the F_0_ level, as well as the absence of any [CuL_2_]Br_2_ absorption bands in the range of the chlorophyll fluorescence emission spectrum (not shown), the assumption that [CuL_2_]Br_2_ acts as a fluorescence quencher seems unlikely.

(c) If [CuL_2_]Br_2_ acted as a very efficient exogenous electron acceptor, then (1) the dark decay of F_V_ reflecting the rate of reoxidation of reduced Q_A_ could be greatly accelerated; (2) the photoinduced growth rate of F_M_ could be decreased; (3) we could observe light-induced oxygen evolution in the absence of exogenous electron acceptors; (4) there could be spectral changes in the absorption spectrum of [CuL_2_]Br_2_, indicating the formation of the reduced form of this agent. However, none of the abovementioned occurs (Figure 2 and Figure 3). So [CuL_2_]Br_2_ does not act as an electron acceptor.

(b) It could be assumed that [CuL_2_]Br_2_ is incorporated somewhere on the donor side or between its components and blocks and/or slows down normal electron donation and/or acts as an exogenous “poor” electron donor to the RC, unable to provide efficient photoaccumulation of reduced Q_A_. However, the lack of slowdown in the rate of F_M_ rise in the presence of [CuL_2_]Br_2_ (Figure 3 kinetics 2–4), and especially the inability of exogenous electron donors to overcome the inhibition by [CuL_2_]Br_2_ and thus restore F_M_ to its original level (not shown) is the evidence against the fact that the donor side is the site of the inhibitory action of this agent. In addition, the indirect data on the redox properties of [CuL_2_]Br_2_ discussed above also testify against this assumption.

(a) Thus, the main site of the inhibitory effect of [CuL_2_]Br_2_ most likely is the components of the PSII reaction center. This assumption is quite convincingly confirmed by all the considerations discussed above, which reject other sites of action of this agent on PSII. In addition, as it was considered in the introduction, the most likely and the most experimentally substantiated main site of action among several auxiliary sites of action of both Cu^2+^ aqua ions and organometallic complexes based on Cu^2+^ seems to be components of the PSII reaction center.

The calculated IC_50_ (30 µM) at a concentration of PSII membranes of 20 µg mL^−1^ chlorophyll is quite comparable with the data obtained by other authors. The value(s) IC_50_ of the PSII oxygen evolution activity inhibition by CuCl_2_ (1) in spinach chloroplasts at a Chl concentration of 20 µg mL^−1^ was 18–20 µM [36]; in the presence of DCBQ or ferricyanide, the values were, respectively, 6.6 μM and 20.7 μM at Chl concentrations of 6.66 μg mL^−1^ or 19 μg mL^−1^ in the PSII membranes and thylakoids, respectively [39]; in the PSII membranes at a Chl concentration of 20 µg mL^−1^, it was 8–10 µM [41]; in the PSII membranes at a Chl concentration of 10 µg mL^−1^, it was 7 µM [88]. The inhibitory activity of diaqua-(N-pyruvidene-β-alaninato) copper(II) monohydrate (Cu(pyr-β-ala)) expressed by the IC_50_ value, as compared with the control samples, was 136 µM at a chlorophyll concentration equal to 30 µg mL^−1^ [15]. The Ki (16 µM), –logKi (4.8), and xt equals one [CuL_2_]Br_2_ molecule on one chlorophyll molecule, or 220–250 inhibitor molecules per one PSII reaction, are different from the data. DCMU: 1 DCMU molecule per around 300 molecules of chlorophyll [67,68], but comparable with the data for a potent inhibitor of photosynthesis, 4,6-dinitro-o-cresol (DNOC): 1 binding site per 2.3 chlorophyll molecules [68]. The values of Ki for the inhibition of oxygen evolution activity by Cu^2+^ aqua ions (as CuCl_2_) or DCMU in the presence of 0.5 mM DCBQ as an artificial electron acceptor obtained by Yruela et al. in 1992 [38] were 5.88 (5.32) µM and 0.023 µM, respectively. As for the [CuL_2_]Br_2_ binding site, the data that [CuL_2_]Br_2_ independently binds with only one binding site per acceptor molecule are a similar conclusion that was drawn for the inhibitor, 4.6-dinitro-o-cresol [68].

Comparing the effects of DCMU and [CuL_2_]Br_2_ on the interaction of PSI with PSII (Figure 7A), it can be roughly estimated that the [CuL_2_]Br_2_ effect is no more than 5–10% of the DCMU effect. Comparing the effects of [CuL_2_]Br_2_ on F_M_ (50%) and on the interaction of PSI with PSII (see above), it is reasonable to propose that the latter effect is probably additional and not main. The existence of additional sites (effects) to the main known that is assumed and experimentally shown for many inhibitors of photosynthesis [6,70], including the aqua ions of Cu^2+^ [50,89]. Using different methods, the additional effects of diuron (1,1-dimethyl, 3-(3’,4’-dichlorophenyl) urea) [90,91,92,93,94], dinoseb (6-sec-butyl-2,4-dinitrophenol) [58,95], and other phenolic TNP inhibitor (2,4,6-trinitrophenol) [96] on the donor side of PSII, have been noted. DBMIB (2,5-dibromo-3-methyl-6-isopropylbenzoquinone), as a strong quencher of the excited state of chlorophyll in the PSII antenna [97], has been identified. Moreover, different authors often indicate completely different loci as the main loci of all of the identified binding sites (effects and mechanisms of action) for the same inhibitor. The dominance of any effects (both on the donor and acceptor side of PSII) depends on the concentration of both the inhibitor and the biological sample [67].

The absence of any change in the AAA fluorescence spectra in the presence of [CuL_2_]Br_2_ compared to the control (including [CuL_2_]Br_2_ concentrations at which the photoinduced electron transfer in PSII is almost completely suppressed, especially during prolonged incubation in the presence of [CuL_2_]Br_2_) indicates that (1) neither the whole [CuL_2_]Br_2_ complex nor the copper ion that could be released from this complex had no significant effect on the AAA PSII proteins native state; (2) and, therefore, there evidently is no release of the copper ion from the [CuL_2_]Br_2_ complex. Previously, it was shown that a number of water-soluble organometallic complexes based on Cu(II) effectively quench the intrinsic fluorescence of AAA of PSII proteins. Based on these and other data, it was proved that copper ions are released from these complexes and bind to tryptophan and tyrosine, thus inhibiting the functioning of the components of the PSII donor side [15]. These complexes, in their case, served as a kind of means of delivering copper ions to the site of action. This is one of the most likely common tasks and roles of the corresponding organic ligands in the composition of organometallic complexes. The combination of metal ions with the corresponding organic ligands makes it possible to obtain complexes that not only maintain the initial biological activity of the metal ions but also have high solubility in an appropriate media and the ability to deliver the metal ions to the site of its action practically in the initial biological activity state.

On the basis of the experimental data obtained, it could be assumed that the site of action of [CuL_2_]Br_2_ is a component of the PSII reaction centre. These data are consistent with the fact that Cu(II) aqua ions also act on PSII RC components [45]. At first glance, it is not easy to imagine how a relatively large [CuL_2_]Br_2_ molecule can penetrate into RC components located in a relatively densely packed and hydrophobic RC region. However, on the one hand, [CuL_2_]Br_2_ itself is a highly hydrophobic compound (see above), and, on the other hand, it has been theoretically substantiated and experimentally shown that the RC as a whole and its individual protein components undergo numerous conformational changes in the process of functioning, as well as at different reversible and irreversible impacts [98,99,100,101,102,103,104,105]. Taking these data into account, the assumption that the RC components are the object of [CuL_2_]Br_2_ inhibition seems to be quite realistic. The data on the absence of quenching the intrinsic fluorescence of AAA by [CuL_2_]Br_2_ do not in the least contradict the assumption that the site of action of [CuL_2_]Br_2_ is RC. There are circular dichroism data that indicate that even in the case of a significant quenching of AAA intrinsic fluorescence by Cu^2+^ aqua ions due to their binding to a red fluorescent protein (DsRed) or to its derivatives, no effect of the Cu^2+^-binding on the secondary structure or on the conformation of these proteins was revealed [106].

## 5. Conclusions

According to the purpose of the work, to investigate the possible effects of a new organometallic complex, (CuL_2_]Br_2_) PSII, (1) several different interesting effects were identified and studied at the level of different PSII regions. (2) The assumption was confirmed that the presence of a ligand significantly changes the possible site and mechanism of the inhibitory action of copper ions. The experimental data at the time of writing this article can be explained by the fact that a possible mechanism of inhibition could be some conformational changes in the structure of the reaction center due to its interaction with the inhibitory agent, especially during photoaccumulation. To elucidate the exact mechanism and site of the inhibitory action of this agent, it is planned to significantly expand the list of research methods and the types of biological samples and apply the method of stabilizing the structure of the reaction center. The results obtained can be used in the development of methods for the guaranteed delivery of the active element to the site of its main activity in the organic complex.

## Figures and Tables

**Figure 1 cells-11-02680-f001:**
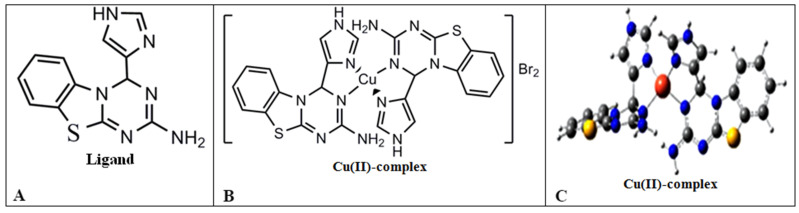
Tested ligand (4H-1,3,5-triazino [2,1-b]benzothiazole-2-amine,4-(2-imidazole)) (**A**) and Cu(II)-complex (Bis{4H-1,3,5-triazino [2,1-b]benzothiazole-2-amine,4-(2-imidazole)}copper(II) bromide) (**B**,**C**). (**A**,**B**), expanded structural formulas. (**C**), optimized molecular structures.

**Figure 2 cells-11-02680-f002:**
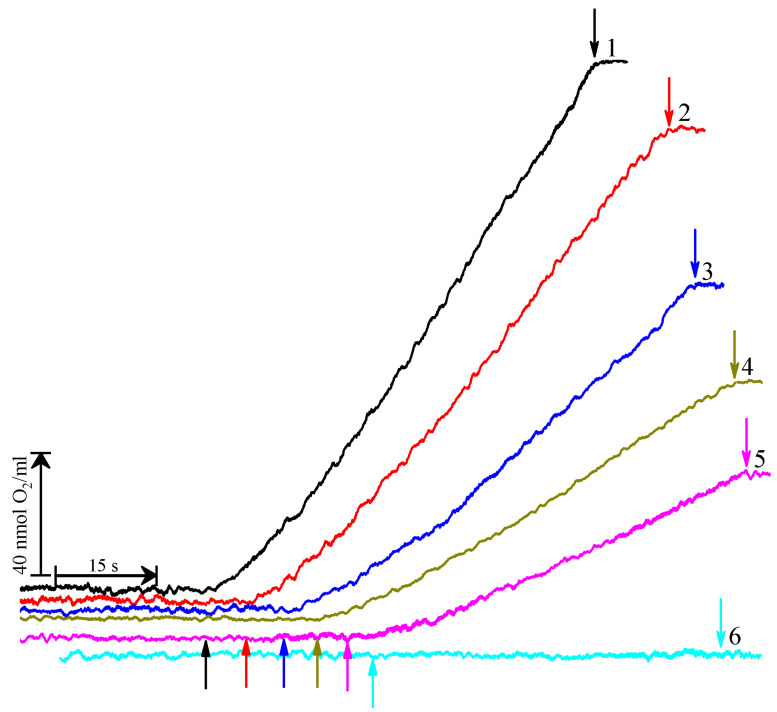
Kinetics of the PSII membranes oxygen evolution in the absence of any additions (1) and in the presence of [CuL_2_]Br_2_ at concentration of 3 µM (2); 20 µM (3); 50 µM (4); 100 µM (5). The assay medium contained 50 mM MES (pH 6.5), 0.33 M sucrose, 15 mM NaCl, 0.1 mM DCBQ, 1 mM FeCy. ↑ and ↓—light (λ = 650 nm, 1200 μmol photon s^−1^ m^−2^) on and light off, respectively. The kinetic 6 was measured in the absence of artificial electron acceptors but in the presence of 30 μM [CuL_2_]Br_2_. The fact that PSII-membranes evolve no oxygen at this condition testifies that [CuL_2_]Br_2_ can’t serve as artificial electron acceptor. The PSII membranes concentration was equivalent to 20 µg mL^–1^ of chlorophyll.

**Figure 3 cells-11-02680-f003:**
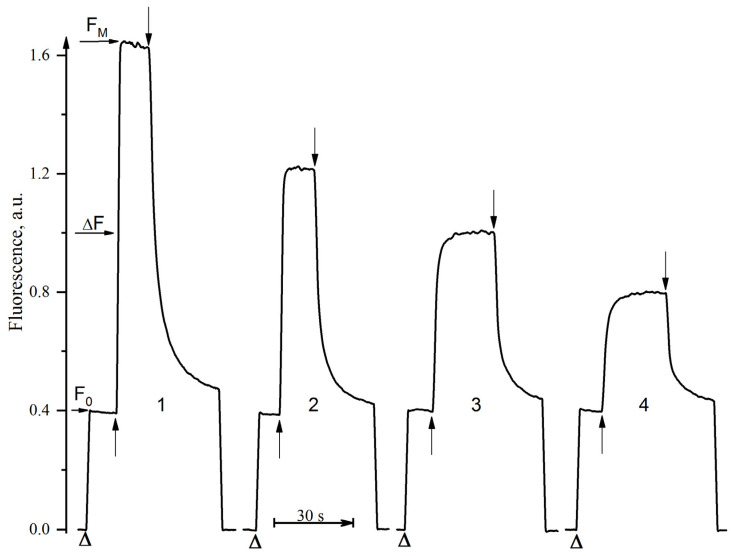
Kinetics of the photoinduced changes of the PSII chlorophyll fluorescence yield (ΔF) related to the photoreduction of the primary quinone electron acceptor, Q_A_, in PSII membranes in the absence of any additions (kinetics 1) and in the presence of [CuL_2_]Br_2_ at concentration of 8 µM (kinetics 2), 30 µM (kinetics 3) and 100 µM (kinetics 4). Triangle symbol indicates the moment of switching on the measuring light (λ = 490 nm, 4 μmol photons m^−2^ s^−1^) exciting PSII chlorophyll fluorescence, F_0_ (λ ≥ 650 nm). The upward ↑ and downward ↓ arrows indicate the moment of respective switching on and off the actinic light (λ > 600 nm, 1000 μmol photons m^−2^ s^−1^). The PSII membranes concentration was equivalent to 20 µg mL^–1^ of chlorophyll.

**Figure 4 cells-11-02680-f004:**
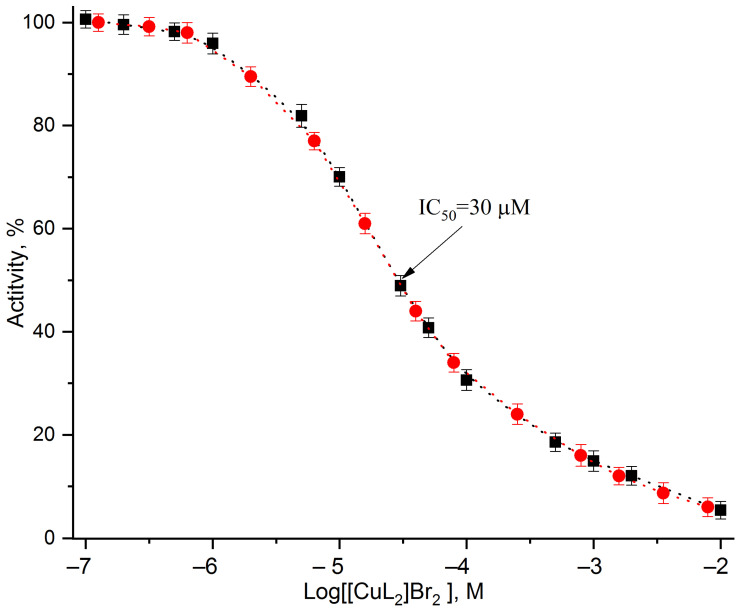
Dependence of the PSII membranes photochemical activity inhibition on [CuL_2_]Br_2_ concentration, measured both as the photosynthetic oxygen evolution (black squares) and as light-induced changes of the PSII chlorophyll fluorescence yield (red circles). Data from four independent experiments are represented as mean ± SD. The PSII membranes concentration was equivalent to 20 µg mL^–1^ of chlorophyll.

**Figure 5 cells-11-02680-f005:**
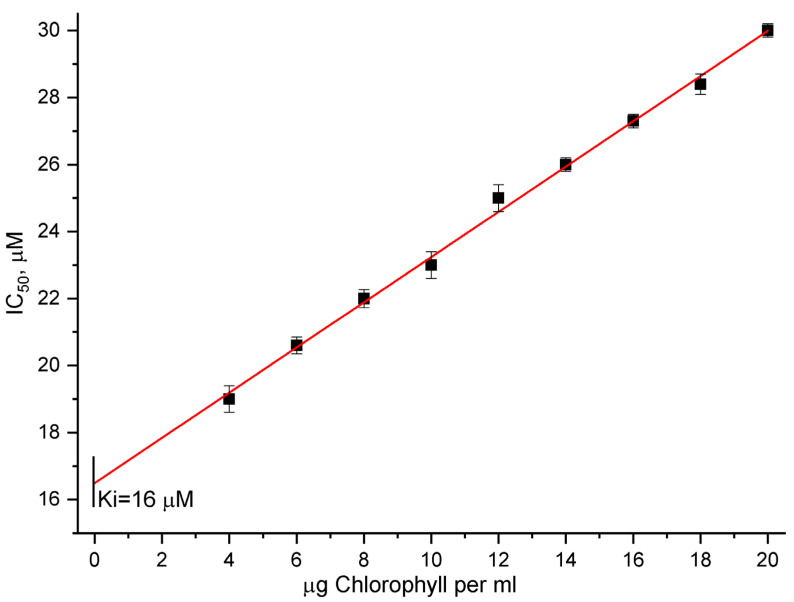
Changes of I_50_ values of the PSII oxygen evolution inhibition by various [CuL_2_]Br_2_ concentrations in dependence on PSII membrane concentrations used in the experiments expressed as concentrations of chlorophyll. Reaction medium and measurement conditions are described in the legend of Figure 1.

**Figure 6 cells-11-02680-f006:**
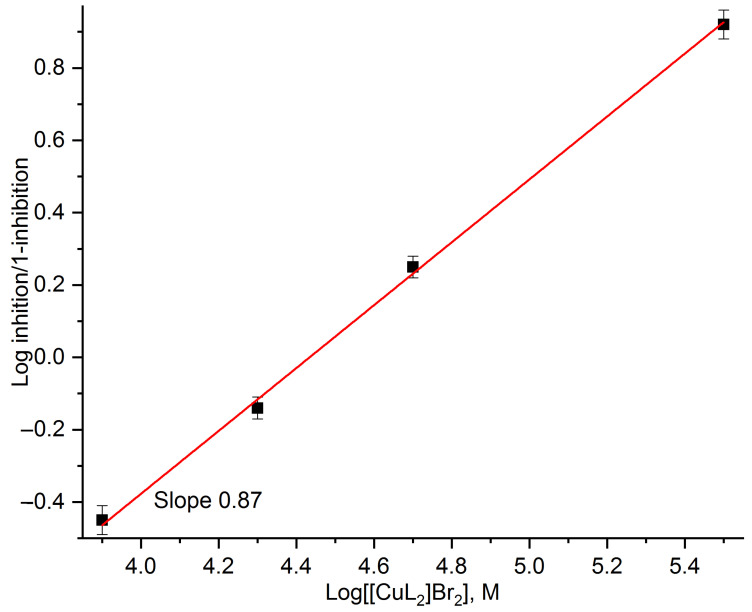
Dependence of Log(inhibition/(1-inhibition)) on Log([CuL_2_]Br_2_ concentration) (Hill plot) calculated from the inhibition data of the PSII light-induced oxygen evolution by various [CuL_2_]Br_2_ concentrations. Reaction medium and measurement conditions are described in the legend of Figure 1.

**Figure 7 cells-11-02680-f007:**
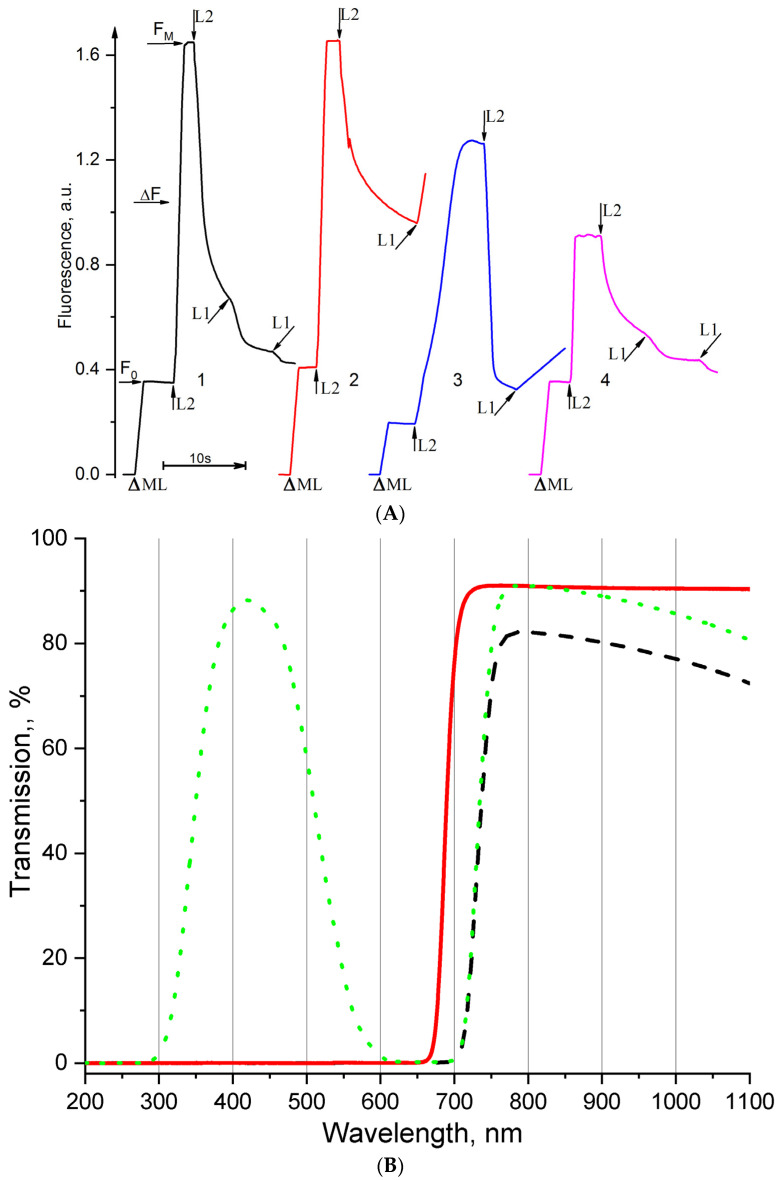
(**A**). Kinetics of the photoinduced changes of the PSII chlorophyll fluorescence yield (ΔF) related to the photoreduction of the primary quinone electron acceptor, Q_A_, in spinach thylakoids in the absence of any additions (kinetics 1) and in the presence of 0.5 µM DCMU (kinetics 2), 7 µM DBMIB (kinetics 3) and 100 µM [CuL_2_]Br_2_ (kinetics 4). Triangle symbol indicates the moment of switching on the measuring light (λ = 490 nm, 4 μmol photons m^−2^ s^−1^), exciting PSII chlorophyll fluorescence, F_0_ (λ ≥ 650 nm). The upward and downward arrows with L2 indicate the moment of respective switching on and off the actinic Light-2 (L2), exciting both photosystems (λ > 600 nm, 1000 μmol photons m^−2^ s^−1^). Arrows pointing up at an angle and down at an angle with L1 indicate the moment of respective switching on and off so-called Light 1 (L1), exciting predominantly PSI (λ > 707 nm, 800 μmol photons m^−2^ s^−1^). The spinach thylakoids concentration was equivalent to 20 µg mL^–1^ of chlorophyll. (**B**). Transmission spectrum of so-called Light 1 (L1) light, exciting predominantly PSI (shown by black dash line), was obtained by combination of glass light filers SZS20 (shown by green dot line) plus KS19 (shown by red line).

**Figure 8 cells-11-02680-f008:**
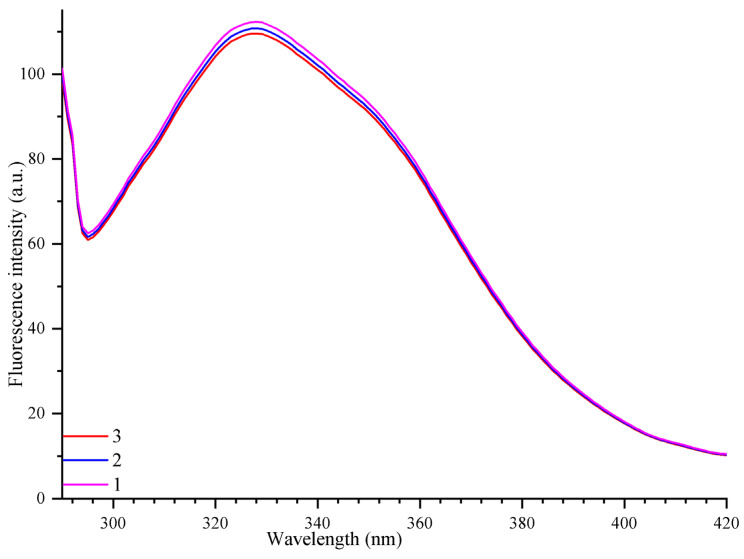
Emission spectra of the aromatic amino acids intrinsic fluorescence (mainly Trp, Tyr) of components of the PSII proteins in the PSII membranes suspension without (1) and with 30 µM [CuL_2_]Br_2_ (2, 3) measured after the PSII membranes incubation with [CuL_2_]Br_2_ in the dark for 3 min (2) and 21 min (3). Excitation wavelength 275 nm. Fluorescence emission was recorded in the range from 290 nm to 420 nm. The DMSO concentration in all samples was the same as in the control (not more than 1%). The PSII membranes concentration was equivalent to 10 µg mL^−1^ of chlorophyll.

## Data Availability

The data presented in this study are available on request from the corresponding authors.

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
