# Peer review of "Effects of Novel Photosynthetic Inhibitor [CuL2]Br2 Complex on Photosystem II Activity in Spinach"

_cells, 2022, doi:10.3390/cells11172680_

Round 1

Reviewer 1 Report

This paper reports on the synthesis and basic characterization of a novel photosystem II inhibitor molecule ([CuL2]Br2 complex , L= 21 bis{4H-1,3,5-triazino[2,1-b]benzothiazole-2-amine,4-(2-imidazole).

The data are of interest and worth publishing. The authors concluded (792-793) that " it could be assumed that the site of action of [CuL2]Br2 is the components of the PSII reaction centre".  

I agree with this assumption, and think that it would be important to confirm this assumption. Thus, I suggest to perform another simple experiment (the authors might have already performed): to determine the proportion of active reaction centers (RCs) remaining in the presence of [CuL2]Br2 complex. This can be determined by measuring Fv in the presence of DCMU. This experiment could show if (i) the complex eliminated the ability of a fraction of RCs to generate stable charge separation, while having no effect on the remaining PSII RC population; or (ii) there is a partial inhibition affecting the activity of each RC.

Alternatively, the authors may want to present thermoluminescence data in the presence and absence of DCMU; deltaA515 absorbance transients on thylakoid membranes might also be of interest - which would, at the same time, provide information on the effect of the complex membrane permeability.

A critical remark on Fv/Fm: this value cannot and should not be equated with the "PSII maximum quantum photochemical yield" (see Sipka et al. 2021 Plant Cell (ref 103), and references therein). Also, strictly speaking, FM,  does not necessarily "characterizes the efficiency of electron supply from the donor-side components to QA through the PSII reaction center (RC)", while  this parameter (and Fv/Fm) certainly can be used to monitor the photochemical activity and the associated structural dynamics of PSII.  Please correct all relevant parts.

Minor, mostly technical remarks:

(i) Abbreviation DCBQ should be introduced when first used (l. 119).

(ii) FM or Fm, FO or F0.

(iii) Corrigenda: an PSII -- a PSII (26); charges -- charge (122); mg Chl-1 -- (mg Chl)-1 (165); restored -- reduced (206); Calc. (?) (247); ll 253-257: add missing spaces; registrated -- recorded (536); electron rate -- electron transfer rate (607); This is not possible with PSII-membranes, as in our case. -- rephrase, unclea (608); fact tesifies -- facts testify (627); Comparising -- Comparing (729); strongly -- strong (739); medias -- media (761); shown -- have shown (784).

Reviewer 2 Report

The manuscript submitted in the journal "Cells" by Zharmukhamedov et al., entitled "Effects of novel photosynthetic inhibitor [CuL2]Br2 complex on photosystem II activity in Spinach" is interesting for the scientists working on different aspects of photosynthesis. 

I found the manuscript worthy of publication though not before the language modifications that appears problematic at some places and some kay points elaborated and essentially required as mentioned below:

Abstract: Line # 25: remove 'an' before PSII

Line # 28: Remove 'the' before 'both'

Line # 35: rephrase "concentration coincided almost completely"

Introduction:

Line # 51:   Please remove unnecessary words and rephrase " the main reactions that are typically found in plant cells only and probably could be safer for humans and fauna".

Lines 53 - 61: May be rephrased and summarized in few lines as the text is about the general concepts on PSII and need to be elaborated in so lengthy paragraph.

Line # 67: Remove 'there were' after mainly PSII and add 'have been made' before 'to develop new inhibitory.....'

Lines 72 - 83: Please look for the grammatical errors in the entire paragraph

Lines 109 - 120: While some of the self explanatory abbreviatins are already mentioned in the text , it is suggested to include full form of 'DCMU' and 'DCBQ'.

Materials and Methods:

The reference # 51 "Khorobrykh and Ivanov" explains the isolation of thylakoid membrane but not the conditions of a green house and details on 'how plants were raised?' Please elaborate the growing conditions with light intensity, temperatures etc., which are very important!

Discussion:

Lines 277 - 305: Please go thorough grammatical errors with the help of a native english speaker. Long sentences do not warranty a clear understanding on what is being elaborated?

Similar is the case for explanation on Figure 4 at page 9 (lines 380 onwards).

Conclusions drawn are not up to the mark where the outcome of research and novelty should be discussed regarding [CuL2] Br2 instead of repeating what was investigated under what aims and objectives. Needs significant improvement.

Round 2

Reviewer 1 Report

The authors answered my critical remarks and made the corrections I requested; also provided a short outlook concerning the continuation of this work.